# RUL prediction method based on sequential health index evaluation with multidimensional coupled degradation data

Feng Han[1,2]*, Bo Mo[1]

1 School of Aerospace Engineering, Beijing Institution of Technology, Beijing, China, 2 Beijing Aerospace Automatic Control Institute, Beijing, China

* 3120225045@bit.edu.cn

## Abstract

Remaining Useful Life (RUL) prediction is crucial for implementing predictive maintenance strategies, however, RUL prediction is severely constrained by the lack of high-quality labeled life-cycle data. Moreover, complex coupling relationships exist within the obtained multidimensional degradation data, making it difficult to construct an accurate health index (HI) for the system. To address this challenge, we propose an RUL prediction method based on sequential healthy index evaluation which incorporate two parts: the parameter prediction process and the health index fusion process. The core innovation of this study is an RUL prediction method that integrates a CNN-Transformer hybrid model with a sequential health index evaluation scheme. Compared to traditional data-driven methods, our approach incorporates a chunk-interaction mechanism into the multi-head attention design, thereby reducing model complexity and computational demands. Simultaneously, the sequential evaluation scheme dynamically constructs the health index based on the Mahalanobis distance and the Sequential Evaluation Ratio (SER), which eliminates the reliance on high-quality labeled life-cycle data. Experimental results demonstrate that the proposed method outperforms existing deep learning approaches (such as LSTM, Transformer, and Att-BiGRU) across multiple datasets, exhibiting higher prediction accuracy and robustness, particularly in label-scarce scenarios.

## 1 Introduction

Predicting the Remaining Useful Life (RUL) of complex systems is a crucial component of Prognostics and Health Management (PHM) [1,2] and predictive maintenance strategies [3]. However, accurate RUL prediction is highly dependent on the availability of labeled life-cycle data. Furthermore, as systems become increasingly integrated, multidimensional degradation parameters often exhibit complex coupling characteristics [4]. Relying on a single degradation parameter for prediction tends to

**Data availability statement:** "The data we used are all from a public dataset called Commercial Modular Aero-Propulsion System Simulation (C-MAPSS) dataset—a publicly available aircraft engine performance degradation dataset provided by NASA's Prognostics Center of Excellence. The primary repository is the NASA Prognostics Data Repository hosted by the NASA Ames Research Center. The dataset can be directly downloaded from the following URL: https://ti.arc.nasa.gov/tech/dash/groups/pcoe/prognostic-data-repository/#turbofan The specific dataset files are named train_FD001.txt, test_FD001.txt, RUL_FD001.txt, etc., corresponding to the four sub-datasets (FD001-FD004) used in our study.".

**Funding:** The author(s) received no specific funding for this work.

**Competing interests:** The authors have declared that no competing interests exist.

overlook the interdependencies among multiple parameters, thereby preventing an accurate overall RUL estimation [5].

Currently, data-driven RUL prediction methods are widely used due to their strong adaptability and generalization capability in practical applications [6]. Data-driven approaches can be broadly classified into machine learning-based methods and various hybrid fusion methods [7]. Machine learning methods encompass deep learning [8], with common models including Support Vector Machines (SVM), Gaussian Process Regression (GPR), Convolutional Neural Networks (CNN), recurrent neural networks (RNNs), and Transformers. Recent advances include: Li et al. [9] predicting turbine engine RUL with LS-SVM; Shen et al. [10] introducing intermediate-domain SVM for bearings; Zheng et al. [11] using dilated CNNs for motors; Rathore et al. [12] employing attention-based Bi-LSTM for bearings. However, machine learning methods often have a black-box nature, leading to a lack of model interpretability. To enhance interpretability and accuracy, various fusion strategies have been developed. Chen et al. [13] combining RNNs with Wiener processes; Ma et al. [14] applying PSO-optimized neural networks; Chen et al. [15] integrating CNN-LSTM with feature selection. Such fusion approaches significantly improve both performance and interpretability. In recent years, with the deepening understanding of complex coupling relationships between system components, Graph Neural Networks (GNNs) have been introduced into the RUL prediction field, demonstrating unique advantages. GNNs can explicitly model the topological relationships between sensors or subsystems, treating multidimensional degradation data as graph structures for processing. For instance, the DCAGGCN model captures dynamic dependencies between components through a Dynamic Causal Attention Graph Convolutional Network [16]; DyWave-BiAGCN combines dynamic wavelet transforms with a bidirectional attention mechanism to simultaneously capture time-frequency domain features and global dependencies [17]. These methods have achieved outstanding performance in systems with strong coupling characteristics (such as aero-engines and complex mechanical systems), providing new ideas for handling multidimensional coupled degradation data.

Another issue in data-driven lifetime prediction is the complex coupling relationships within multidimensional degradation data. For systems with simple functions, a single degradation parameter can directly reflect performance degradation. However, for the complex system, multidimensional degradation parameters often collectively encapsulate the system's RUL information; relying solely on a single degradation parameter cannot yield comprehensive results. The data collection process for complex systems is typically costly and technically challenging, making it difficult to obtain high-quality, labeled full life-cycle data in practice. Therefore, a deep exploration of the underlying correlations among historical degradation parameters and the rational construction of a Health Index (HI) are key to enhancing RUL prediction accuracy. To address this issue, this study proposes a dynamic sequential evaluation – based prediction method capable of accurately predicting the system's degradation states and achieving lifetime prediction without life labels. This method adopts the concept of parameter prediction followed by index fusion to indirectly achieve lifetime prediction.

It utilizes a CNN-Transformer model to discover the hidden correlations among degradation parameters and capture their dynamic variations. Through the Sequential Evaluation Ratio (SER), it quantifies deviations in the system's health states to construct health index curves for subsystems. These indexes are then fused via a comprehensive evaluation metric to derive the system's health index, reflecting its degradation state.

Compared to traditional data-driven methods, the CNN-Transformer model proposed in this study tackles the challenge of modeling the coupled relationships among multidimensional degradation parameters by integrating CNN's local feature extraction capability with Transformer's global temporal dependency modeling. Its innovations are reflected in: (1) introducing a chunk-interaction mechanism and Multi-Head Latent Attention (MLA), significantly reducing computational complexity while maintaining long-sequence prediction capability; (2) employing a sequential health index evaluation scheme that dynamically quantifies system state deviation, eliminating the reliance on lifecycle labels required by traditional methods. Compared to existing deep learning approaches (such as LSTM, Transformer, and Att-BiGRU), our method demonstrates significant advantages in model lightweighting, multidimensional data fusion, and adaptability in label-scarce scenarios, providing a feasible solution for real-time predictive maintenance in industrial settings.

## 2 Methodology

### 2.1 Problem formulation

To address the complex nonlinear relationships among degradation parameters and the varying impact of each subsystem's degradation on the overall system health, the proposed method is structured as follows: First, the HI based on the degradation parameters of each subsystem is constructed. Then, the future values of these subsystem HIs are predicted. Finally, the system's overall HI and its RUL are derived based on the criticality of the subsystems and a comprehensive evaluation scheme.

Assuming the system has $K$ key degradation parameters, with the degradation parameter sequence $X_k = \{x_{k,1}, x_{k,2},..., x_{k,t}\}$, where $k = 1,2,...,K$. By analyzing each parameter, the health index $h_k(t)$ for its corresponding subsystem can be constructed, which characterizes the degree of degradation of that subsystem at time point $t$. Mathematically, the health index for a subsystem can be expressed as:

$$h_k(t) = g_k(X_k(t))$$

(1)

where $g_k$ represents the health index construction function of the subsystem, and $X_k(t)$ represents the degradation parameter of the subsystem at time point $t$. To capture the nonlinear correlations among subsystems and achieve accurate temporal prediction of the HI, a prediction function $f$ is trained on historical multidimensional health index sequences. This function is used to predict future HI values. Sliding time windows are applied to the historical data, forming a high-dimensional feature matrix that encapsulates temporal information. The process is shown in Fig 1. and can be expressed in the following form:

$$H(t) = [h(t-(L-1)); h(t-(L-2)); ...; h(t-1); h(t)]$$

(2)

The HIs of multiple subsystems are integrated through a subsystem evaluation scheme to obtain a comprehensive system health index $HI(t)$. The fusion function $\varphi$ incorporates the importance of the subsystems and the evaluation scheme, and is mathematically expressed as:

$$HI(t) = \varphi(\hat{h}_1(t), \hat{h}_2(t), ..., \hat{h}_K(t); \omega_1, \omega_2, ..., \omega_K)$$

(3)

where $\omega_k$ is the comprehensive weight coefficient for the subsystems, satisfying the constraint $\sum_{k=1}^{K} \omega_k = 1$. The fusion operator $\varphi$ is selected based on the system's structure and degradation characteristics; common choices include linear weighting or extreme value operators. Finally, the RUL is determined based on the first-passage time.

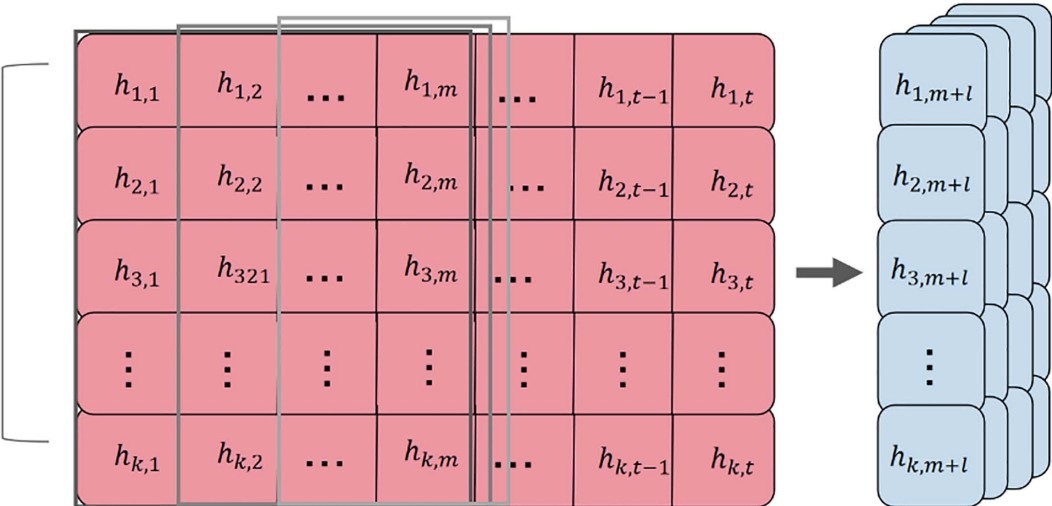

**Fig 1. Schematic diagram of sample slicing for multidimensional historical health index sequence data.**

## 2.2 Framework of the sequential health index evaluation algorithm

This study adopts a two-stage workflow: first performing time-series prediction, and then constructing the health index. The overall framework of the proposed method is illustrated in Fig 2. The lifetime prediction method is based on a CNN-Transformer model. A dynamic sequential evaluation method is introduced to derive the subsystem HI by quantifying the deviation between the observed system state and a predefined healthy state. These subsystems' HIs are then fused using a comprehensive evaluation metric to obtain the overall system health index. This design enables the model to meet common industrial resource constraints (e.g., limited GPU memory and computational power) and real-time requirements (e.g., fast inference on streaming data).

The CNN-Transformer model is designed to capture both local and global dependencies in multidimensional degradation data, making it adaptable to various degradation modes. The CNN module extracts spatial correlations between parameters through convolutional kernels, while the Transformer module with MLA captures long-term temporal dependencies. This combination allows the model to handle complex coupling relationships, even in scenarios with conflicting degradation patterns (e.g., when subsystems exhibit opposite trends). To address potential conflict issues during the Health Index (HI) fusion process, the model optimizes the weight coefficients $\omega_k$ and the fusion operator $\varphi$ based on the importance and degradation characteristics of the subsystems. The fusion function $\varphi$ integrates the health indices of the individual subsystems into a comprehensive health index, while fully considering potential coupling conflicts that may arise from inconsistent degradation trends among the subsystems. The weight coefficients $\omega_k$ are allocated based on the importance and degradation consistency of the subsystems, thereby balancing conflicting trends. For instance, if the health index of a particular subsystem significantly deviates from others (indicating a conflict), its weight can be dynamically adjusted using the Analytic Hierarchy Process (AHP) to reduce its impact on the overall health index. Furthermore, the fusion operator (such as the linear weighting operator) helps mitigate conflicts by reinforcing the information from subsystems whose degradation trends are consistent with the overall trend. As shown in Section 3.3, this method ensures that the fused health index maintains monotonicity and robustness even in scenarios with conflicting parameter coupling.

In the proposed framework, the CNN serves as a relational feature extraction module, tasked with capturing the coupling relationships among the system's multidimensional degradation parameters [18]. To enhance the stability of the

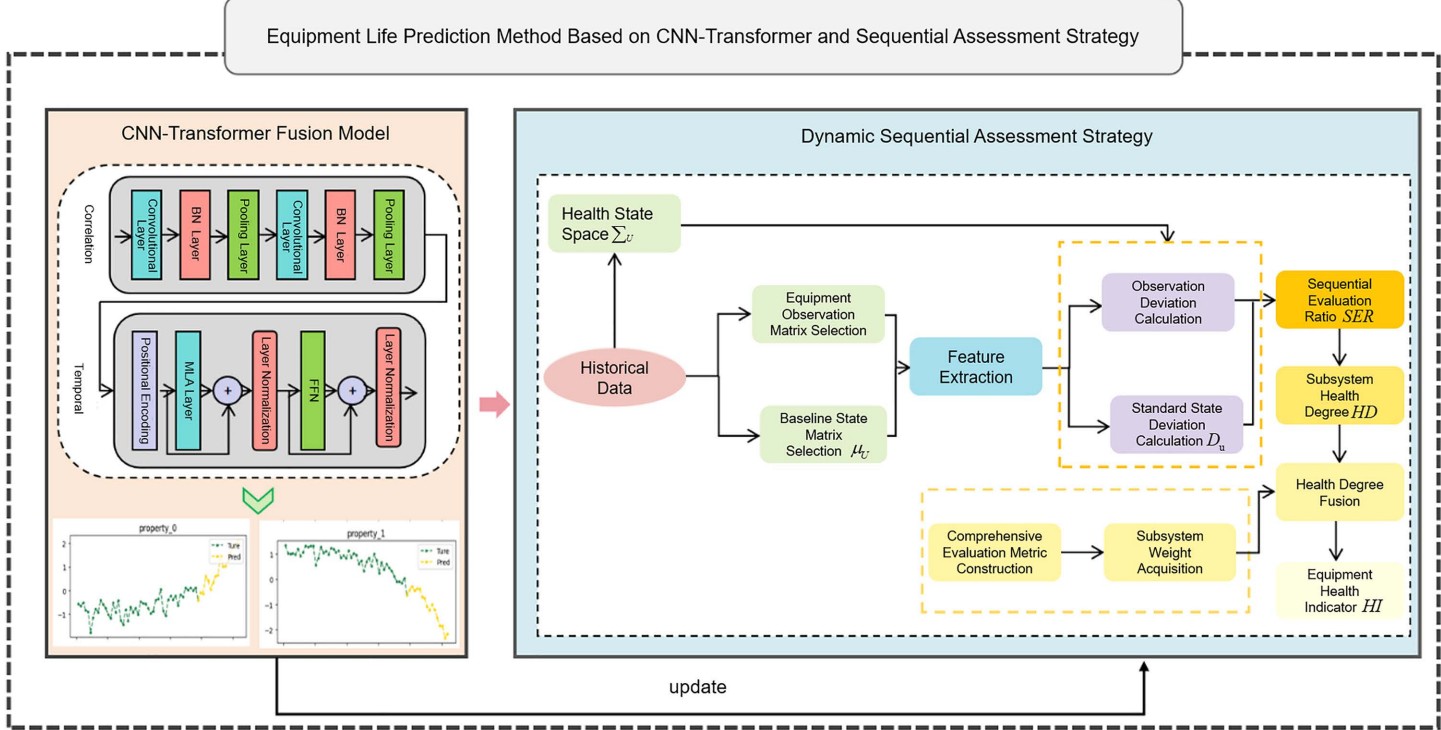

**Fig 2. Framework of the CNN-transformer model and sequential evaluation strategy for RUL prediction.**

feature distribution, a Batch Normalization (BN) layer is incorporated into the CNN, which constrains gradient updates within the non-saturation linear region and improves the model's generalization capability.

Concurrently, the Transformer [19] is adopted as the backbone of the temporal feature extraction module. To address the requirements for reduced computational complexity and practical deployment inherent in lifetime prediction tasks, a Multi-Head Latent Attention (MLA) mechanism is proposed to replace the standard Transformer encoder.

However, the computational complexity of the core self-attention mechanism in the standard Transformer architecture is $O(L2)$, which imposes a significant burden for long-sequence lifetime prediction tasks. To reduce computational complexity for model lightweighting and practical deployment, a common strategy is to adopt a chunking mechanism. This involves partitioning the long sequence into multiple non-overlapping chunks and computing attention independently within each chunk. However, this chunk-based processing can lead to "global information loss." Specifically, partitioning a sequence creates hard boundaries between chunks. This means that the hidden state of a timestep at the end of chunk A cannot directly interact with another critical timestep at the beginning of chunk B via the attention mechanism. Since degradation processes are often continuous and long-term, crucial degradation features may be distributed across different chunks. This inter-chunk isolation prevents the model from capturing long-range dependencies across chunk boundaries, leading to a loss of global sequence coherence and ultimately compromising long-term prediction accuracy.

To address this issue, a cross-chunk interaction mechanism is introduced. This mechanism captures and propagates global dependencies between chunks through the use of global tokens. Specifically, a learnable global token is added to each chunk. This token interacts with all local tokens within its chunk via the attention mechanism, thereby aggregating the chunk's local information. Subsequently, these global tokens from different chunks interact with each other through a lightweight cross-chunk attention layer, enabling the flow and integration of global information. This design effectively

establishes "information bridges" between independent chunks, successfully mitigating the global information fragmentation problem caused by chunking without significantly increasing computational complexity (reduced from $O(L^2)$ to $O(L \cdot P)$, where P is the chunk size). A schematic diagram is shown in Fig 3.

Although GNNs demonstrate excellent performance in processing explicit graph-structured data, their effectiveness highly depends on the quality of the predefined graph structure. For the complex systems targeted by this study (e.g., aero-engines), the precise physical connections or dynamic coupling weights between sensors are often difficult to obtain a priori. In contrast, the CNN-Transformer model adopted in this study implicitly learns local spatial correlations between sensor parameters through the CNN's convolutional kernels, and captures global long-term dependencies through the Transformer's attention mechanism. This approach does not require a predefined graph structure, making it more adaptable to industrial scenarios where sensor relationships are ambiguous or dynamically changing. This "implicit coupling learning" paradigm ensures performance while reducing the model's reliance on prior knowledge, thereby enhancing the method's generalizability.

## 2.3 Sequential HI evaluation strategy

The lifetime prediction method in this study adopts an indirect approach. It requires constructing a reasonable health index that reflects the system's degradation states from multidimensional degradation parameters. The core idea of the proposed sequential evaluation method is to establish a standard state space using historical health data. It calculates the distance between the standard health state feature vector and the observed state feature vector to this standard state space. The ratio of these distances serves as the Sequential Evaluation Ratio (SER). The magnitude of the SER reflects the degree of state deviation, thereby constructing the health index value. This study employs the sequential evaluation method to construct health index curves for the subsystems, which describe the system's degradation process and the

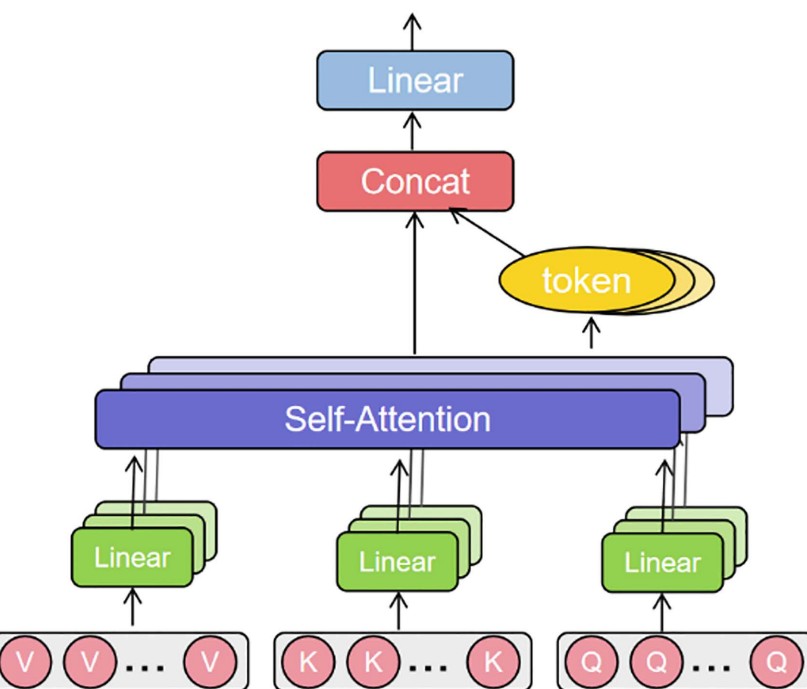

**Fig 3. Schematic diagram of the multi-head latent attention mechanism.**

extent of damage. This enables timely warnings at failure states and facilitates system lifetime prediction. The specific workflow of the sequential evaluation method is shown in Fig 4.

Step 1: Acquire historical data for each subsystem and process the data using the sliding window method to obtain processed data for each subsystem, expressed as follows:

$$X(t) = \begin{bmatrix} x_{1,t-L+1} & x_{1,t-L+2} & \cdots & x_{1,t} \\ x_{1,t-L+1} & x_{1,t-L+2} & \cdots & x_{1,t} \\ \vdots & \vdots & \ddots & \vdots \\ x_{1,t-L+1} & x_{1,t-L+2} & \cdots & x_{1,t} \end{bmatrix} \tag{4}$$

Step 2: Perform time-frequency domain feature extraction on the acquired data to extract data features. Commonly used time-frequency domain features include maximum/minimum values, mean, standard deviation, kurtosis, root mean square (RMS), skewness, spectral energy, and spectral entropy.

Step 3: Obtain the standard health feature vector $\mu_U$. Based on features extracted from the system's healthy state, calculate the mean $\mu$ to derive the standard health feature vector, where $n$ denotes the number of features selected for the system:

$$\mu_U = \begin{bmatrix} \mu_1 & \mu_2 & \mu_3 & \cdots & \mu_n \end{bmatrix} \tag{5}$$

Step 4: Obtain the system's health memory matrix $\sum_U$. Extract features from initial healthy-state data of the system, compute its covariance matrix to highlight correlations between different features and across time steps.

$$\sum_U = \begin{bmatrix} Cov(X_1,X_1) & Cov(X_1,X_2) & \cdots & Cov(X_1,X_n) \\ Cov(X_2,X_1) & Cov(X_2,X_2) & \cdots & Cov(X_2,X_n) \\ \vdots & \vdots & \cdots & \vdots \\ Cov(X_n,X_1) & Cov(X_n,X_2) & \cdots & Cov(X_n,X_n) \end{bmatrix} \tag{6}$$

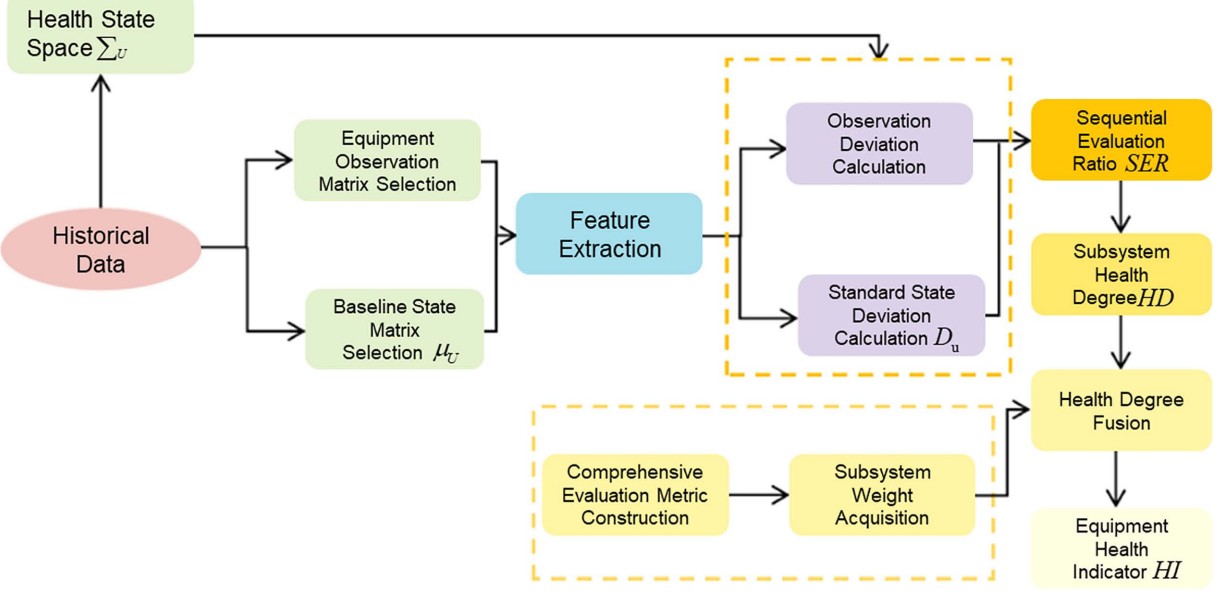

**Fig 4. Workflow diagram of the sequential evaluation method.**

Step 5: Perform Mahalanobis distance calculation to measure the state deviation between the current feature vector of the system and the standard health feature vector. The Mahalanobis distance formula is as follows:

$$D = \sqrt{(x - \mu_U)^T {\sum_U'}^{-1} (x - \mu_U)}$$

(7)

where, $x$ is the feature vector of the system at the current timestep.

Step 6: Calculate the sequential evaluation ratio $SER$. Compute the ratio between the Mahalanobis distance of the system's current state and that of its initial healthy state to obtain the sequential evaluation ratio, which describes the damage state of the system:

$$SER = \frac{D_{current}}{D_{health}}$$

(8)

Step 7: Calculate the health degree $HD$ of the subsystem. The magnitude of the health degree describes the system 's health status, typically constrained between 0 and 1. Generally, values above 0.8 represent a healthy state, while values below 0.4 indicate a warning state.

$$HD = \frac{1}{1 + \alpha \cdot \exp(SER)}$$

(9)

where, $\alpha$ is a tension parameter that controls the influence of the sequential evaluation ratio on the health degree. It can be determined based on expert knowledge or system data monitoring intervals.

where, the tension parameter $\alpha$ is a scalar greater than 0, and its physical significance lies in adjusting the sensitivity of the system's health state to observed deviations. Specifically:

$\alpha$ determines the rate at which the Health Degree HD(t) decreases as the Sequential Evaluation Ratio SER(t) increases.A larger $\alpha$ value indicates that the system is more sensitive to minor state deviations; the health degree would decline rapidly even with a slight increase in SER. This is suitable for systems with extremely high safety requirements that need early warnings.Conversely, a smaller $\alpha$ value indicates a higher tolerance for deviations within the system, resulting in a more gradual decline in health degree. This is suitable for systems with slow degradation processes that allow for a certain buffer period.

This study employs a data-driven grid search approach to determine the optimal $\alpha$ value. The procedure is as follows:

Define the Optimization Objective: The goal is to ensure that the constructed overall system Health Index HI(t) possesses optimal monotonicity and robustness. The calculation methods for these two metrics are described in Section 3.3.

Set the Search Range: A reasonable range (e.g., $\alpha \in [0.5, 5]$) is defined, and a sequence of candidate $\alpha$ values is generated with a fixed step size (e.g., 0.1) over the training set.

Evaluation and Selection: For each candidate $\alpha$, the health index curves for all training units are calculated according to Equations (7)–(9), and fused to obtain the system-level HI. The average monotonicity and robustness of these HI curves are then computed.

Select the Optimal Value: The $\alpha$ value that yields the highest combined score (e.g., monotonicity + robustness) is selected as the final parameter. In the application to the C-MAPSS dataset in this experiment, the optimal $\alpha$ value determined by this method was 1.5.

To quantitatively evaluate the reasonableness of the constructed Health Index (HI), this study adopts a widely recognized set of evaluation metrics, including Correlation, Trendability, Monotonicity, Predictability, and Robustness. The definitions and calculation methods for these metrics are as follows.

(1) Correlation

Correlation(*corr*) is used to measure the similarity of trends in multidimensional sensor data within a complex system, reflecting the relevance of module health degrees at monitoring time points. The Maximal Information Coefficient (MIC), based on the mutual information of all monitoring sensors' data, is employed to quantify the strength of linear or non-linear relationships between two degradation parameters. MIC is a non-parametric correlation measure based on mutual information, whose core idea involves optimizing the mutual information estimate through dynamic grid partitioning. The specific steps are as follows:

Calculate the mutual information: Given two random variables $X$ and $Y$ (referring to two degradation parameters in this context), compute the K-L divergence between their joint distribution and the product of their marginal distributions:

$$I(X;Y) = \int_{XY} P(X,Y) \log \frac{P(X,Y)}{P(X)P(Y)}$$

(10)

Search across different grid partitioning schemes to obtain the maximum normalized mutual information. The calculation is as follows:

$$MIC(X;Y) = \max_{xy<B(n)} \frac{I(X;Y)}{\log(\min(X,Y))}$$

(11)

where $I(X;Y)$ represents the mutual information value between the two degradation parameters; $x$ and $y$ denote the sizes of the partitioning grids; $B(n)$ indicates the maximum number of partitions, typically set to $B(n) = n^{0.6}$. Assuming the equipment has N-dimensional sensor monitoring data, the performance metric $Corr(X)$ describing the trendability of each dimension of sensor monitoring data is obtained. The calculation formula is as follows:

$$Corr(X) = \frac{1}{N-1} \sum_{i=1}^{N-1} |MIC(X;Y_i)|$$

(12)

(2) Trendability

Trendability(*Tre*) refers to the changing trend of the equipment's health degree over time. It is calculated by fitting the trend of the health degree change and determining the slope of the health degree with respect to time, where $t$ represents time.

$$Tre(X) = \frac{\sum_{i=1}^{N} (x_i - x_0) \cdot t_i}{\sum_{i=1}^{N} t_i^2}$$

(13)

(3) Monotonicity

With the accumulation of equipment operating time, component wear occurs, leading to performance degradation. Typically, the deviation of degradation parameters can represent the equipment's degradation extent. Although the influence of noise and changes in operational conditions may cause parameters to exhibit short-term non-monotonic behavior, the long-term trend generally demonstrates monotonicity. This study uses the monotonicity metric to describe the overall degradation trend of equipment performance and the smoothness of the data. The monotonicity metric for equipment monitoring parameters is calculated as follows:

$$Mon\left(X\right) = \frac{\left|\sum_{j=1}^{N-1} \text{sgn}\left(X\left(t_{j+1}\right) - x\left(t_j\right)\right)\right|}{N-1} \tag{14}$$

where $X\left(t_j\right)$ represents the monitoring data of the equipment at time $t_j$, $N$ denotes the total number of sensor monitoring instances, and $\text{sgn}()$ represents the sign function.

(4) Predictability

Predictability refers to the dispersion of sensor data at the time of equipment failure and the range of data variability based on sensor categories. The predictability metric is calculated as follows:

$$Pre\left(X\right) = exp\left(-\frac{std\left(X_{T_f}\right)}{\left|mean\left(X_{T_f}\right) - mean\left(X_{T_s}\right)\right|}\right) \tag{15}$$

where $X_{T_f}$ represents the data characteristics of the monitoring parameters at the time of the equipment's functional failure, and $X_{T_s}$ denotes the baseline data from the normal startup phase of the equipment.

(5) Robustness

The magnitudes of equipment degradation parameters differ, leading to varying degrees of susceptibility to noise. Furthermore, as equipment ages, degradation parameters often become more sensitive to noise. This study uses robustness to represent the sensor's tolerance to random noise and outliers, reflecting the stability of the equipment module's health degree when confronted with data noise and anomalous values. The calculation formula for the robustness metric is as follows:

$$\text{Rob}\left(X\right) = \frac{1}{N}\sum_{j=1}^{N} exp\left(-\left|\frac{X_R\left(t_j\right)}{X_T\left(t_j\right)}\right|\right) \tag{16}$$

where $X_T\left(t_j\right)$ represents the smoothed trend component of the health degree for the equipment module at time $t_j$ under noise-affected conditions, $X_R\left(t_j\right)$ denotes the random component of the health degree for the equipment module at time $t_j$, and $N$ represents the total number of health degree data points for the equipment module. Both the smoothed trend component and the random component are obtained through exponential smoothing, and the calculations for all other metrics are performed using the exponentially smoothed data.

The evaluation metrics, including Correlation and Monotonicity, all range from 0 to 1. As described above, a higher value of an evaluation metric indicates that the degradation parameter contains relatively more information regarding equipment degradation. To construct a more reasonable health indicator for the equipment, it is necessary to comprehensively consider the data characteristics of correlation, trendability, monotonicity, predictability, and robustness when selecting essential degradation parameters. Based on the evaluation results, weighted information fusion is performed to obtain an overall measure of the degradation parameters. The calculation is expressed as follows:

$$J = \omega_1 \text{Tre} + \omega_2 \text{Mon} + \omega_3 \text{Pre} + \omega_4 \text{Rob} + \omega_5 Corr \tag{17}$$

where $\sum_{i=1}^{5} \omega_i = 1$, $J$ represents the comprehensive evaluation metric, and $\omega_i$ denotes the weight coefficients of each evaluation metric. The weights for the evaluation metrics are obtained via the Analytic Hierarchy Process (AHP).

# 3 Experimental verification and result analysis

## 3.1 C-MAPSS dataset

This study focuses on complex equipment characterized by multidimensional degradation parameters. To validate the effectiveness of the proposed method for lifetime prediction, experiments were conducted using the Commercial Modular Aero-Propulsion System Simulation (C-MAPSS) dataset, a publicly available aircraft engine performance degradation dataset provided by NASA's Prognostics Center of Excellence. The system is described below.

As the core component of propulsion systems in jet-powered aircraft [20], turbine engines are widely used in aerospace applications and represent a quintessential example of a complex system with multidimensional degradation parameters. During operation, subsystems such as turbines, compressors, and combustion chambers exhibit intricate interdependencies—for instance, combustion efficiency directly affects turbine performance, which in turn impacts the engine's overall power output [21]. These interactions create deeply coupled relationships among components and subsystems within the engine system. Moreover, acquiring degradation data for such complex system is costly and technically challenging, making high-quality full lifecycle data with direct labels difficult to obtain in practical applications. The NASA Prognostics Center of Excellence developed the Commercial Modular Aero-Propulsion System Simulation (C-MAPSS) dataset for aircraft engine performance degradation. This dataset meets the validation requirements of this study and is a widely used public benchmark in system lifetime prediction research. The C-MAPSS dataset offers high relevance, rich data diversity, and varied experimental conditions. Fig 5. illustrates the turbine engine model alongside its module interconnections and layout schematic.

During the data collection process for the C-MAPSS dataset, high-fidelity engine models were utilized to simulate the degradation process of turbofan engines under various operating conditions. All engines used in the simulation were of the same type. Different failure modes were injected into the engines during the simulation. Additionally, environmental noise interference and sensor errors were incorporated during data acquisition to approximate realistic flight conditions. The dataset records the complete lifecycle data of turbofan engines from normal operational status to failure state, encompassing data from 4 subsets. Each record consists of the engine unit ID, the number of operational cycles (where one complete engine run from take-off to landing is considered one operational cycle), 3 operational setting parameters, and 21 sensor measurements.

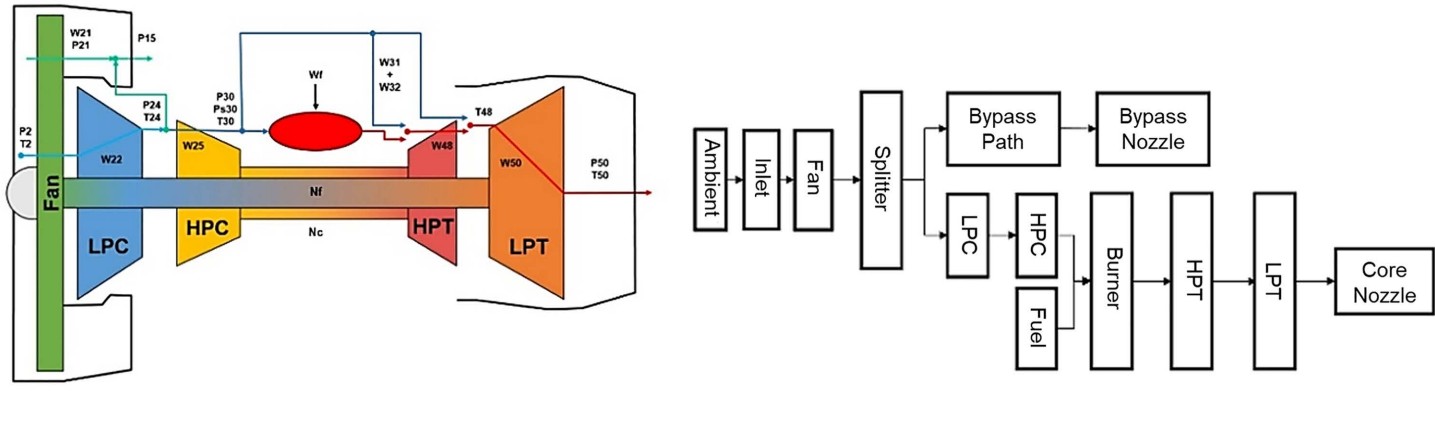

(a) Model Schematic (b)Interconnection Diagram

**Fig 5. C-MAPSS turbofan engine model schematic and module interconnection diagram.**

 

A detailed description of the dataset is presented in Table 1. This table includes the number of engines in the training set, the number of distinct operational condition settings (i.e., combinations of operating parameters), the number of failure modes, and the maximum number of operational cycles per engine. The injected degradation corresponds to two failure modes: High-Pressure Compressor (HPC) degradation and fan degradation. Subsets FD001 and FD003 contain only HPC degradation, whereas FD002 and FD004 include both failure modes. The training set data comprises the complete lifecycle data of the engines. In contrast, the test set contains truncated operational sequences that end at some point before failure. During data acquisition, although the engines were of the same type, their initial state was an unknown non-failure state. This reflects the uncertainties arising from manufacturing variations and component differences encountered in practical scenarios. In this study, the 21 condition monitoring variables are utilized as degradation parameters for experimentation. These parameters primarily consist of data related to temperatures, pressures, rotational speeds, etc., from various engine modules. Based on their recording sequence within the dataset, these parameters are designated as sm_X, where X indicates the recording order of the monitoring variable. The specific meaning represented by each sm_X parameter is not reiterated here.

## 3.2 Data preprocessing

During the data preprocessing stage, effective denoising is crucial for enhancing the robustness of subsequent model predictions. Common time series data denoising methods include wavelet transform, moving average, Kalman filtering, and exponential smoothing. Although wavelet transform can effectively handle abrupt changes in non-stationary signals, it requires the selection of appropriate wavelet bases and decomposition levels, leading to relatively high complexity. The moving average method is simple but can easily lead to phase lag and excessive trend smoothing.

This study ultimately selected exponential smoothing for denoising, primarily based on the following considerations:

(1) Data Characteristics Matching: The sensor degradation data in the C-MAPSS dataset typically exhibits a relatively smooth gradual process rather than containing a large number of high-frequency abrupt changes. Exponential smoothing, by assigning higher weights to recent data, effectively preserves this slow degradation trend while suppressing random noise.

(2) Computational Efficiency and Simplicity: The exponential smoothing method is computationally simple, requires no complex parameter tuning (such as wavelet base selection), and has very low computational overhead. This aligns well with the overall goals of lightweight design and industrial applicability pursued by this method.

(3) Synergy with the Prediction Model: The core of this research is multi-step time series prediction. Exponential smoothing itself is a fundamental time series forecasting method. The data preprocessed by it shares an inherent conceptual consistency with the subsequent Transformer-based time series prediction model, as both emphasize smooth sequence evolution and temporal dependencies.

To evaluate the effectiveness of exponential smoothing in this study, we compared it with a typical wavelet denoising method (using 'db4' wavelet, soft thresholding, 3-level decomposition). On the FD001 subset, we trained and tested the

**Table 1. Specific introduction to the dataset.**

|  | FD001 | FD002 | FD003 | FD004 |
|---|---|---|---|---|
| Engine units for training | 100 | 260 | 100 | 249 |
| Engine units for testing | 100 | 259 | 100 | 248 |
| Operating conditions | 1 | 6 | 1 | 6 |
| Fault modes | 1 | 1 | 2 | 2 |
| Maximum Number of Operational Cycles | 362 | 378 | 526 | 544 |

same CNN-Transformer model architecture using the original data, wavelet-denoised data, and exponentially smoothed data, respectively. Using RMSE as the primary evaluation metric, the results are shown in Table 2.

Experimental results show that for sensor data with smooth degradation characteristics, such as the C-MAPSS dataset, exponential smoothing yields slightly better prediction accuracy than wavelet denoising. This supports our decision to select it as the preferred denoising scheme. Furthermore, its lower implementation complexity and computational cost make it more suitable for efficiency-oriented industrial prediction scenarios.

To validate the accuracy of the proposed fusion model based on the Multi-Head Latent Attention mechanism and the dynamic sequential Evaluation strategy, the C-MAPSS degradation dataset was adopted to verify the performance of the prediction method and the reasonableness of the health curve fusion approach. Given the temporal correlation inherent in the degradation parameters and the presence of environmental noise interference during system time-series data acquisition, Exponential Smoothing (ES) was modified for the smoothing pre-processing of multidimensional degradation parameters. The calculation is expressed as follows:

$$\begin{cases} y_0 = x_0 & , t = 0 \\ y_t = r \sum_{i=0}^{t} (1-r)^i x_{t-i}, t > 0 \end{cases}$$

(18)

where $r$ represents the decay coefficient, $x$ represents the raw data of the degradation parameters, and $y$ represents the pre-processed value of the degradation parameter. Through experimental investigation in this study, the decay factor $r$ was set to 0.3. This value achieves an effective balance between noise removal (smoothing effect) and preservation of the degradation trend: an $r$ value that is too small (e.g., 0.1) leads to excessive smoothing, potentially obscuring early degradation features; while an $r$ value that is too large (e.g., 0.5) results in insufficient filtering. The pre-processed value $y_t$ at the current time is a weighted result of historical data with unequal weights, exhibiting stronger correlation with adjacent time points. This demonstrates that the method can smooth the degradation parameters while preserving their inherent variation trends. Performance parameters collected by different sensors exhibit varying dimensions. Prior to parameter prediction and health curve construction, monitoring parameters with zero variance were excluded. The remaining multi-dimensional data were then standardized. Consequently, all data in subsequent processing stages are dimensionless. Fig 6. displays the visualization results of data before and after denoising for Engine No. 9 in the FD001 subset.

Furthermore, for the FD002 and FD004 subsets, the combinations of three operational settings are more numerous, and operational data under different settings exhibit significant variations. To mitigate the impact of working conditions on method validation results, data screening was performed on these two subsets. Based on the values of the operational altitude parameter os_3, data with os_3 ≈ 100 (matching the operational conditions of the FD001 and FD003 subsets) were selected. Fig 7. presents the operational data for Engine No. 2 in the FD002 subset. After data filtering, the degradation trends of parameters such as sm_13 can be visually identified.

After the complete data preprocessing pipeline (including exponential smoothing denoising, removal of zero-variance parameters, data standardization, and data filtering for subsets FD002 and FD004 based on the os_3 condition), the final dataset for model training and testing was obtained. The statistics of the filtered data scale are shown in Table 3.

**Table 2. Impact of different denoising methods on prediction performance (FD001).**

| Denoising Method | RMSE (Mean) | Remarks |
|---|---|---|
| Raw Data | 0.1258 | Contains significant noise, adversely affecting the model's ability to learn the true degradation pattern. |
| Wavelet Denoising | 0.1154 | Effectively reduces noise but may introduce minor distortions or over-smooth trends. |
| Exponential Smoothing (r = 0.3) | 0.1116 | Achieves the best overall performance on this dataset. |

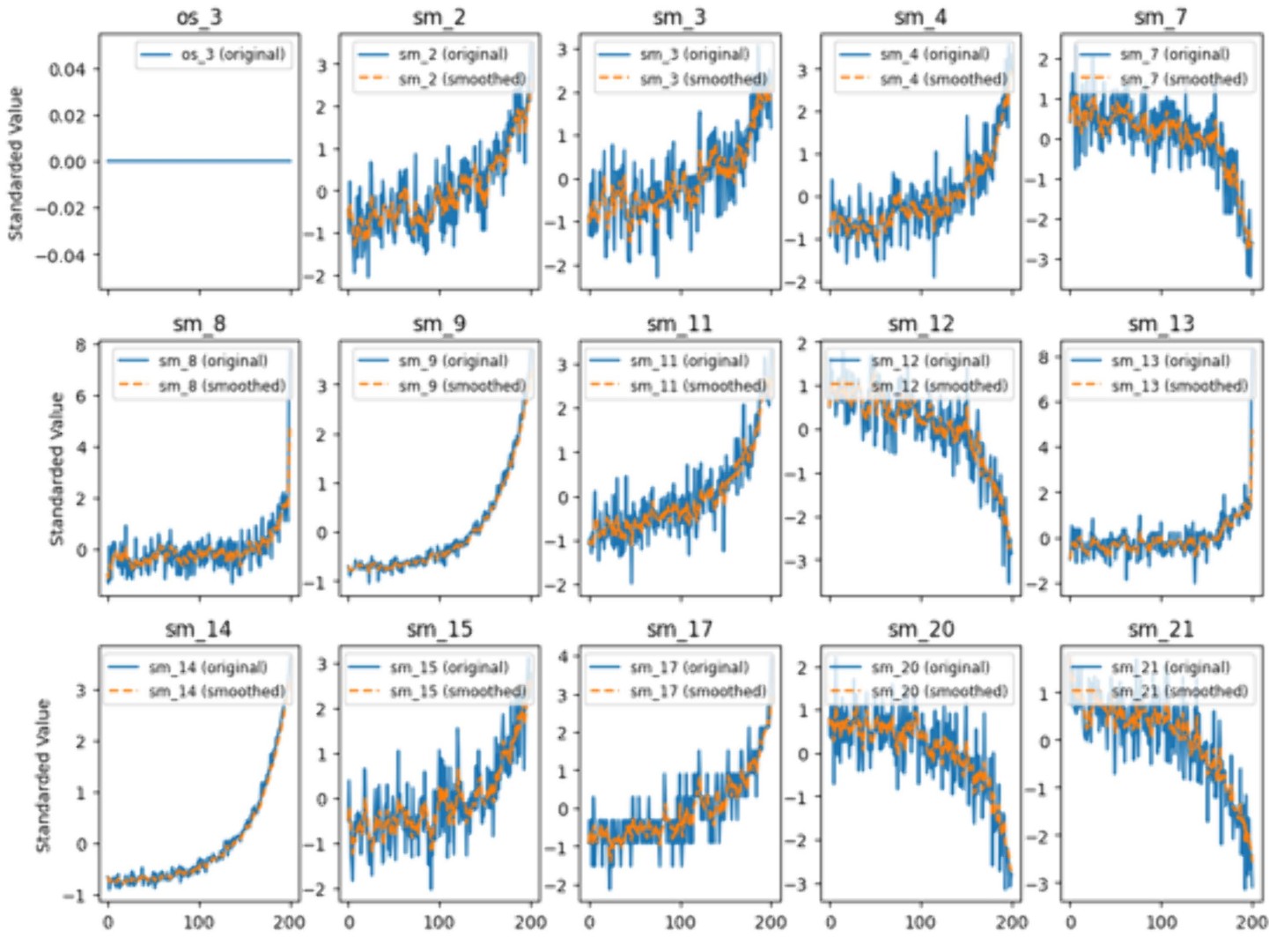

**Fig 6. Data visualization for engine No. 9 in the FD001 dataset.**

As shown in Table 3, for the FD001 and FD003 subsets (which have only one operating condition), all data were retained. For the FD002 and FD004 subsets, to control for the operating condition variable, we filtered data where os_3 ≈ 100, ultimately retaining approximately 57% of the engine units. All subsequent reported experimental results are based on this filtered dataset.

### 3.3 Experimental results

This study employs the C-MAPSS degradation dataset to validate the overall predictive efficacy of the proposed CNN-Transformer fusion model and sequential evaluation strategy. Root Mean Square Error (RMSE), Mean Absolute Error (MAE), and the Coefficient of Determination ($R^2$) are adopted as evaluation metrics. These metrics analyze the discrepancy between the ground-truth health index and the predicted health index, thus enabling an indirect evaluation of the accuracy of system lifetime prediction. During experimentation, data segmentation was implemented using a sliding window approach with a window length of 16 and step size of 1. The selection of this window length was based on the

**Fig 7. Data visualization for engine No. 2 in the FD002 dataset.**

**Table 3. Effective data size of each subset after data preprocessing.**

| Subset | Original Training Engines | Filtered Training Engines | Retention Ratio | Original Testing Engines | Filtered Testing Engines | Retention Ratio |
|--------|--------------------------|---------------------------|-----------------|--------------------------|--------------------------|-----------------|
| FD001 | 100 | 100 | 100% | 100 | 100 | 100% |
| FD002 | 260 | 150 | 57.7% | 259 | 150 | 57.9% |
| FD004 | 249 | 140 | 56.2% | 248 | 140 | 56.5% |

following considerations: In the C-MAPSS dataset, one operational cycle represents a complete flight mission. Setting the window length to 16 cycles ensures coverage of a sufficiently long continuous operational phase to capture the short-term dynamic patterns of the degradation process and the coupled relationships between parameters. Simultaneously, this length achieves a balance between computational efficiency and information completeness: excessively short windows fail to provide adequate temporal context, while overly long windows would significantly increase the model's computational

burden and potentially introduce irrelevant early historical information. K-fold cross-validation (K = 5) was adopted to ensure result stability and generalization capability. For each fold, the network prediction model was trained with the Adam optimizer, Mean Squared Error (MSE) loss function, a learning rate of 0.05, batch size of 256, and maximum training epochs of 500. An early stopping mechanism was incorporated: validation was performed every 5 epochs; training terminated prematurely if validation loss showed no improvement over 5 consecutive evaluations, with optimal model parameters retained to prevent overfitting and conserve computational resources.

The proposed CNN-Transformer fusion model consists of two components. First, a Convolutional Neural Network (CNN) serves as a correlative feature extractor to capture interdependencies among multi-dimensional parameters; specifically, the input feature dimension is 17, with the CNN employing 2D convolutions using 4 kernels to process feature dimensions and yielding a 12-dimensional output. Second, the temporal feature module adopts a Transformer architecture incorporating chunk and cross-chunk interaction mechanisms within its multi-head attention computation to achieve model light weighting, configured with 6 encoder layers, 4 multi-head attention heads, a chunk size of 2, and a feedforward network comprising two linear layers (64 neural units per layer, ReLU activation), resulting in a final 14-dimensional output. Experimental comparisons evaluate the proposed method against Att-BiGRU, Transformer, WDCNN, and LSTM approaches.

During experimentation, tests were conducted on individual system units using a prediction horizon equivalent to 30% of their operational cycles. The mean and variance of evaluation metrics were calculated, with comparative results shown in Fig 8. Detailed quantitative outcomes are presented in Tables 4–6. Experimental results demonstrate that the proposed fusion model outperforms comparative methods in temporal prediction performance. Across all data subsets, the model achieves $R^2$ consistently exceeding 0.8 and maintains RMSE below 0.1. Furthermore, lower metric variances indicate enhanced stability and reliability, attributable to: (1) the CNN's capacity for extracting multidimensional parameter correlations, (2) the Transformer's long-term forecasting superiority, and (3) reduced model parameters/computational complexity through the improved multi-head potential attention mechanism.

Experimental results demonstrate that the proposed method outperforms comparative methods across RMSE, MAE, and $R^2$ metrics, exhibiting particularly outstanding performance on multi-operating-condition datasets such as FD002 and

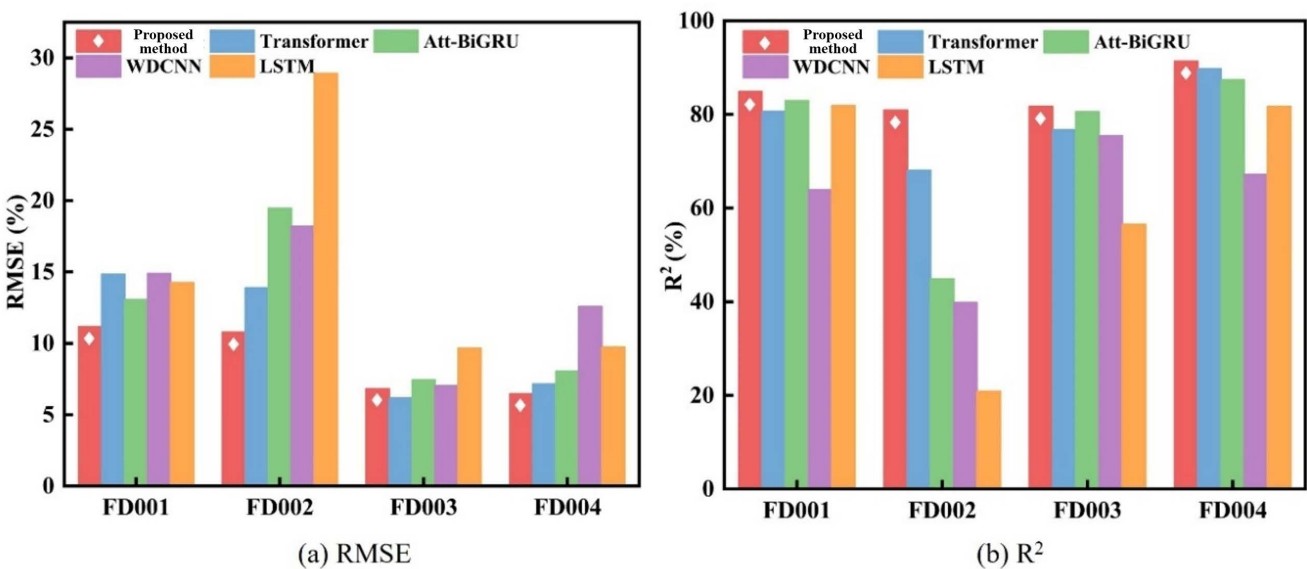

**Fig 8. Comparison of RMSE and $R^2$ results.**

**Table 4. RMSE results comparison.**

| | RMSE | Proposed method | Transformer | Att-BiGRU | WDCNN | LSTM |
|---|---|---|---|---|---|---|
| FD001 | Ave | 0.1116 | 0.1483 | 0.1306 | 0.149 | 0.1426 |
| | Std | 0.0529 | 0.073 | 0.0741 | 0.0777 | 0.0741 |
| FD002 | Ave | 0.1077 | 0.139 | 0.1947 | 0.182 | 0.2891 |
| | Std | 0.0142 | 0.0314 | 0.0245 | 0.0523 | 0.0771 |
| FD003 | Ave | 0.0681 | 0.0617 | 0.0744 | 0.0703 | 0.0966 |
| | Std | 0.0359 | 0.0205 | 0.0391 | 0.0271 | 0.0368 |
| FD004 | Ave | 0.0645 | 0.0715 | 0.0804 | 0.1257 | 0.0973 |
| | Std | 0.0193 | 0.0184 | 0.035 | 0.0218 | 0.0248 |

**Table 5. $R^2$ results comparison.**

| | $R^2$ | Proposed method | Transformer | Att-BiGRU | WDCNN | LSTM |
|---|---|---|---|---|---|---|
| FD001 | Ave | 0.8486 | 0.8067 | 0.8293 | 0.6392 | 0.8188 |
| | Std | 0.0862 | 0.1733 | 0.1590 | 0.5402 | 0.1596 |
| FD002 | Ave | 0.8090 | 0.6806 | 0.4488 | 0.3979 | 0.2087 |
| | Std | 0.1194 | 0.2114 | 0.1239 | 0.3877 | 0.4455 |
| FD003 | Ave | 0.8169 | 0.7672 | 0.8058 | 0.7544 | 0.5650 |
| | Std | 0.1386 | 0.2445 | 0.1371 | 0.1884 | 0.3821 |
| FD004 | Ave | 0.9137 | 0.8973 | 0.8738 | 0.6719 | 0.8167 |
| | Std | 0.0308 | 0.0376 | 0.0671 | 0.1297 | 0.0776 |

**Table 6. MAE results comparison.**

| | MAE | Proposed method | Transformer | Att-BiGRU | WDCNN | LSTM |
|---|---|---|---|---|---|---|
| FD001 | Ave | 0.0398 | 0.0474 | 0.044 | 0.0487 | 0.0442 |
| | Std | 0.0169 | 0.0172 | 0.0165 | 0.0174 | 0.0162 |
| FD002 | Ave | 0.0441 | 0.0589 | 0.0792 | 0.0756 | 0.1259 |
| | Std | 0.0052 | 0.0141 | 0.0121 | 0.0229 | 0.0326 |
| FD003 | Ave | 0.0281 | 0.0259 | 0.0299 | 0.0297 | 0.0402 |
| | Std | 0.0159 | 0.0089 | 0.0165 | 0.0119 | 0.0155 |
| FD004 | Ave | 0.025 | 0.0281 | 0.0333 | 0.0549 | 0.041 |
| | Std | 0.0079 | 0.0068 | 0.013 | 0.0108 | 0.0092 |

FD004. Across all data subsets, the model presented in this chapter consistently achieves $R^2$ values above 0.8, while maintaining RMSE metrics at a low level of approximately 0.1. This is attributed to the ability of the CNN-Transformer model to effectively capture coupling relationships among multidimensional parameters, combined with the sequential evaluation strategy's capability to dynamically extract degradation features. Compared to traditional deep learning methods, the proposed approach reduces computational resource demands through lightweight design (e.g., the MLA mechanism), while simultaneously enhancing practicality in real industrial scenarios through label-free health index construction.

To validate the applicability of the proposed method in industrial scenarios, we conducted a quantitative evaluation of the computational efficiency of the proposed CNN-Transformer model and the comparative models. Experiments were performed on a platform equipped with an NVIDIA GeForce RTX 3080 GPU and an Intel i7-11700K CPU, using the PyTorch framework. We measured three key metrics: (1) Model Parameters, reflecting model size and memory footprint;

(2) Floating Point Operations (FLOPs), reflecting computational complexity; and (3) Average Time for a Single Forward Inference (Inference Time), measured with a batch size of 1 to simulate online prediction scenarios. The results are presented in the Table 7.

To comprehensively evaluate the performance of the proposed method, we supplemented comparative experiments with two advanced graph neural network models: DCAGGCN and DyWave-BiAGCN. Since the original GNN models require a predefined graph structure, we constructed adjacency matrices in two ways: (1) a knowledge graph based on the physical connections of the equipment; (2) a data-driven graph based on the correlation coefficients of the sensor data. The experimental results are shown in Table 8.

The results indicate that the CNN-Transformer fusion model proposed in this study achieves prediction accuracy that is superior or comparable to advanced GNN models, while demonstrating significant advantages in terms of model complexity and inference efficiency. This validates that, even without complete knowledge of the system's internal physical connections, the approach of implicitly learning coupled relationships can effectively capture the intrinsic correlations among multidimensional degradation parameters. Furthermore, it proves more suitable for industrial scenarios with high real-time requirements.

After prediction, the sequential evaluation strategy constructs the system's health curve, prioritizing monotonicity and trend characteristics during weight allocation. The constructed HI were evaluated using early/late time consistency, correlation, robustness, and monotonicity metrics, compared against a residual evaluation method without sequential principles. Dynamic thresholds of 0.75 (early) and 0.4 (late) were applied for time consistency calculations, with values <0.4 indicating a warning state. Validation shows consistently high consistency metrics, confirming the strategy effectively extracts degradation features, constructs rational health curves, and accurately characterizes degradation states. Metric results are shown in Table 9.

To validate the model's capability in capturing parameter coupling under different degradation modes, we compared its performance on subsets with single failure modes (FD001 and FD003) and mixed failure modes (FD002 and FD004). As shown in Tables 4–6, the proposed method maintains high $R^2$ (>0.8) and low RMSE (~0.1) across all subsets, demonstrating robustness to varying degradation patterns. Specifically, in FD002 and FD004 (with multiple failure modes), the model effectively captures coupled relationships without significant performance drop, indicating its adaptability to complex degradation scenarios.

**Table 7. Comparison of model computational efficiency.**

| Model | Parameters (M) | FLOPs (G) | Inference Time (ms) |
|---|---|---|---|
| Proposed (CNN-Transformer) | 2.1 | 0.38 | 4.5 |
| Transformer | 4.8 | 0.95 | 9.8 |
| Att-BiGRU | 3.5 | 0.72 | 7.1 |
| WDCNN | 5.2 | 1.10 | 11.3 |
| LSTM | 1.8 | 0.41 | 5.2 |

**Table 8. Performance comparison with graph neural network models (FD001).**

| Model | Graph Construction Method | RMSE | R² | Parameters (M) | Inference Time (ms) |
|---|---|---|---|---|---|
| Proposed (Ours) | Not Applicable | 0.1116 | 0.8486 | 2.1 | 12.8 |
| DCAGGCN | Physical Graph | 0.1189 | 0.8274 | 3.8 | 22.5 |
| DCAGGCN | Data-Driven Graph | 0.1152 | 0.8381 | 3.8 | 22.5 |
| DyWave-BiAGCN | Physical Graph | 0.1203 | 0.8227 | 4.5 | 28.3 |
| DyWave-BiAGCN | Data-Driven Graph | 0.1168 | 0.8335 | 4.5 | 28.3 |

**Table 9. Evaluation results of the sequential HI evaluation strategy.**

| | Sequential Evaluation Strategy | | | | Residual Evaluation Method | | | |
|---|---|---|---|---|---|---|---|---|
| | FD001 | FD002 | FD003 | FD004 | FD001 | FD002 | FD003 | FD004 |
| $Accuracy_{early}$ | 1.00 | 0.93 | 0.99 | 0.98 | 0.39 | 0.51 | 0.31 | 0.73 |
| $Accuracy_{late}$ | 0.92 | 0.75 | 0.94 | 0.83 | 0.46 | 0.92 | 0.96 | 0.75 |
| $mon$ | 0.51 | 0.23 | 0.55 | 0.34 | 0.39 | 0.28 | 0.41 | 0.22 |
| $corr$ | 0.50 | 0.019 | 0.46 | 0.11 | 0.02 | 0.001 | 0.02 | 0.01 |
| $rob$ | 0.37 | 0.36 | 0.36 | 0.37 | 0.36 | 0.32 | 0.36 | 0.26 |

Furthermore, we analyzed the health index fusion in coupling conflict scenarios, where subsystems exhibit contradictory trends. For example, in FD004, some engines show conflicting sensor behaviors (e.g., sm_13 increasing while sm_15 decreasing). The sequential evaluation strategy, combined with weight coefficients ω_k derived from AHP, ensures that the fused HI prioritizes consistent subsystems, mitigating conflicts. This is reflected in the high monotonicity and late-stage consistency metrics (Table 9), confirming the rationality of index fusion even in challenging conditions.

Fig 9 presents health degree distribution histograms for the FD001 and FD003 subsets, comparing the proposed sequential evaluation strategy against the conventional residual evaluation method. Pink solid-line boxes denote statistical results from the proposed method. The figure clearly demonstrates higher differentiation between healthy and warning states, along with enhanced sensitivity to system state transitions. The overall health degree distribution aligns with the ground-truth pattern of progressive degradation over operational time, visually revealing degradation trends and further validating the method's rationality and effectiveness.

To validate the effectiveness of the aforementioned parameter calibration method and demonstrate the impact of the $\alpha$ value on the results, we conducted a sensitivity analysis on the FD001 subset. We compared the key evaluation metrics of the health index curves constructed using different $\alpha$ values ($\alpha = 0.5, 1.5, 3.0$). The results are presented in Table 10.

It can be observed that when $\alpha = 1.5$, the health index curve achieves the best monotonicity and favorable late-stage consistency, which aligns with the objective of our grid search optimization. When $\alpha$ is too small (0.5), although the curve exhibits the highest robustness, its monotonicity decreases significantly, which is unfavorable for characterizing a clear degradation trend. When $\alpha$ is too large (3.0), all metrics show a decline. This experiment demonstrates the necessity of systematically calibrating the parameter $\alpha$ and confirms the rationality of the final selected value of $\alpha = 1.5$.

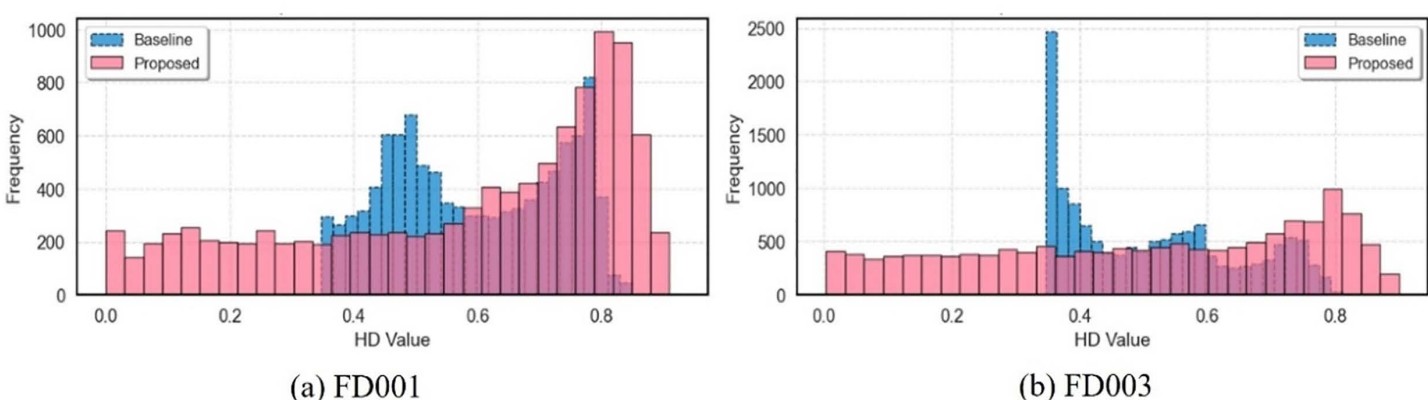

(a) FD001  (b) FD003

**Fig 9. Health index value distribution comparison for FD001 and FD003 subsets.**

Table 10. Impact of different *α* values on health index quality (FD001).

| *α* Value | Monotonicity | Robustness | Early-stage Consistency | Late-stage Consistency |
|---|---|---|---|---|
| 0.5 | 0.42 | 0.55 | 1.00 | 0.85 |
| 1.5 | 0.51 | 0.50 | 1.00 | 0.92 |
| 3.0 | 0.48 | 0.45 | 1.00 | 0.88 |

To visually demonstrate the characterization capability of the HI constructed via the sequential evaluation strategy and comprehensive metric fusion for system degradation, health index curves for all engines in the FD001 and FD003 subsets are presented as heatmaps in Fig 10. The gradual color transitions over time indicate excellent monotonicity and robustness, confirming stable reflection of progressive health degradation. Concurrently, during late operational stages, health indexes consistently fall below the warning threshold (0.4), demonstrating timely and accurate responsiveness to impending failure states.

Compared to direct lifespan prediction methods, the proposed predict-then-fuse approach relies on temporal prediction accuracy, where the precision of multi-dimensional degradation parameter forecasts influences health index construction. To evaluate this impact, single-step predictions generated by the model were used to construct health index curves, which were compared against ground-truth curves in Fig 11. When the prediction model achieves $R^2 \approx 0.9$, the predicted health curves closely align with ground truth, while the HI themselves maintain high $R^2$ values. This confirms that the predict-then-fuse strategy introduces no significant adverse effects when temporal prediction capability is robust, thereby validating the method's effectiveness and rationality.

## 3.4 Ablation study

To quantify the contribution of each core module in the proposed method, we conducted a systematic ablation study. Experiments were performed on the FD001 and FD003 subsets, using RMSE and $R^2$ as evaluation metrics. All ablated models maintained the same hyperparameters as the full model.

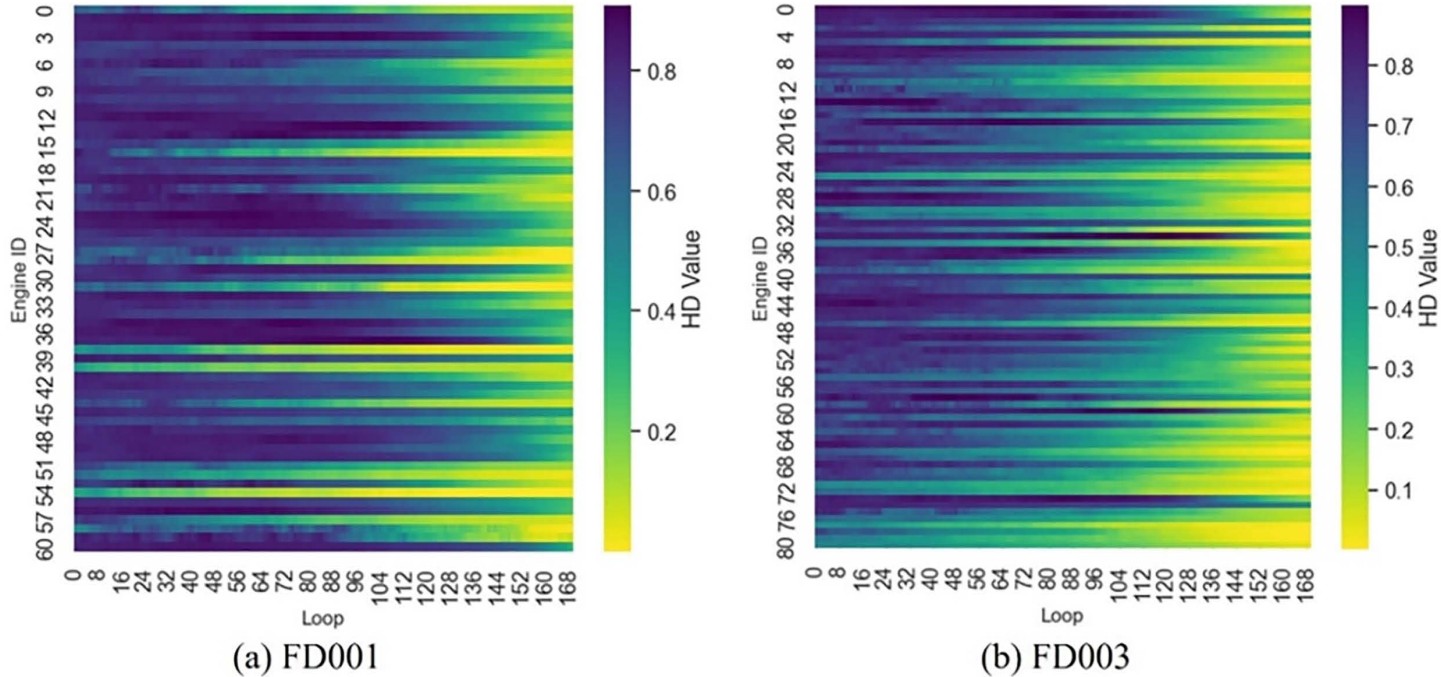

**Fig 10. Health index extraction results for FD001 and FD003.**

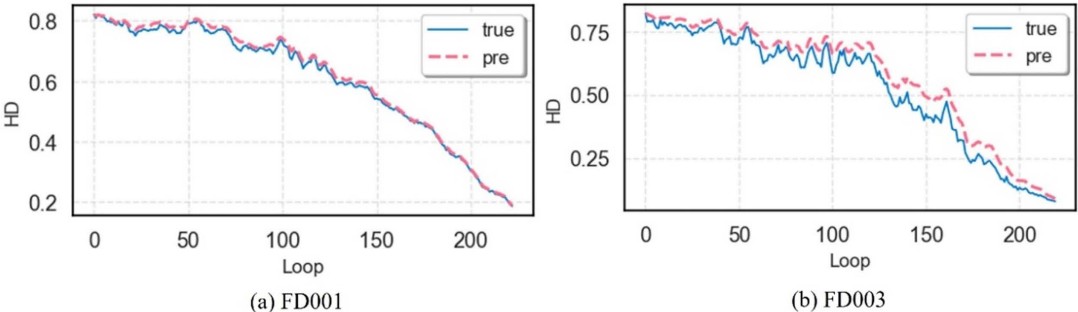

**Fig 11. Health curves for engine No. 9 in FD001 and engine No. 63 in FD003.**

We designed the following model variants to verify the necessity of components in the CNN-Transformer architecture:

Variant A (w/o CNN): Removes the CNN relational feature extraction module and feeds the raw time series data directly into the Transformer. Variant B (w/o MLA): Replaces the Multi-Head Latent Attention (MLA) mechanism with the standard multi-head self-attention mechanism. Variant C (LSTM Replacement): Completely replaces the entire Transformer temporal module with standard LSTM layers. Full Model (Ours): The complete CNN-Transformer model proposed in this study.

The results are shown in the Table 11.

Removing the CNN module (w/o CNN) leads to a noticeable performance degradation, confirming that explicitly modeling the coupling relationships among multidimensional parameters is crucial for accurate prediction. Without the CNN, the model struggles to capture the interactions between subsystems. Removing the MLA mechanism (w/o MLA) also results in performance decline and increased computational complexity. This validates the value of our proposed lightweight attention mechanism in maintaining accuracy while improving efficiency. Replacing the Transformer with LSTM causes the most significant performance loss, highlighting the advantage of the Transformer architecture in capturing long-term temporal dependencies compared to traditional RNN models.

Furthermore, to validate the effectiveness of the sequential health index evaluation strategy, we compared it with two baseline methods: Baseline 1 (Residual) constructs the health index using the traditional model prediction residual-based method [22]. Baseline 2 (Direct Fusion) skips the sequential evaluation and directly constructs the system health index by weighted fusion of raw sensor data. The full strategy (Ours) refers to the SER-based sequential health index evaluation strategy proposed in this study.

Using the same full prediction model and only altering the health index construction method, we evaluated the quality of the final system health index (using the metrics from Table 9). The results are as follows in Table 12.

The proposed sequential evaluation strategy significantly outperforms the two baseline methods in terms of both monotonicity and late-stage consistency. This indicates that dynamically quantifying state deviation using Mahalanobis distance and the SER can more effectively capture the system's degradation trend and provide clear, consistent early warnings upon failure. Baseline 2 (Direct Fusion) exhibits better robustness but extremely poor monotonicity, demonstrating that

**Table 11. Ablation study results on model architectures (mean±standard deviation).**

| Model Variant | FD001 (RMSE) | FD001 (R²) | FD003 (RMSE) | FD003 (R²) |
|---|---|---|---|---|
| w/o CNN | 0.1284±0.061 | 0.821±0.102 | 0.0795±0.041 | 0.781±0.121 |
| w/o MLA | 0.1201±0.058 | 0.835±0.095 | 0.0728±0.038 | 0.802±0.115 |
| LSTM Replacement | 0.1347±0.065 | 0.809±0.110 | 0.0831±0.043 | 0.769±0.129 |
| Full Model (Ours) | 0.1116±0.053 | 0.849±0.086 | 0.0681±0.036 | 0.817±0.139 |

**Table 12. Ablation study results of health index evaluation strategies (FD001).**

| Evaluation Strategy | Monotonicity | Robustness | Late-stage Consistency |
|---|---|---|---|
| Baseline 1 (Residual) | 0.39 | 0.46 | 0.39 |
| Baseline 2 (Direct Fusion) | 0.28 | 0.58 | 0.45 |
| Full Strategy (Ours) | 0.51 | 0.50 | 0.92 |

fusing untreated raw data fails to form a clear health degradation curve. Baseline 1 (Residual method) performs poorly across all metrics, highlighting the superiority of the data-driven sequential evaluation approach in the absence of a precise physical model. In summary, the ablation study compellingly demonstrates that each module within the proposed CNN-Transformer model, as well as the sequential evaluation strategy, is indispensable. They collectively contribute to the overall excellent performance of the method.

## 4 Conclusions

This study addresses the challenge of predicting the Remaining Useful Life (RUL) of complex equipment with multidimensional degradation parameters under unlabeled or label-scarce conditions by proposing a method based on historical degradation data. This method indirectly achieves lifespan prediction by performing temporal prediction of the equipment's multidimensional degradation parameters and constructing health indicator curves. It tackles several challenges, including high data acquisition costs, the difficulty of obtaining full life-cycle data, the high complexity of existing methods, and their impediment to practical deployment. To this end, this study proposes an RUL prediction method based on a CNN-Transformer and sequential health index evaluation. Its core innovations include: achieving model lightweighting through a chunk-interaction mechanism and Multi-Head Latent Attention (MLA), significantly reducing computational complexity; and dynamically constructing the health index using Mahalanobis distance and the Sequential Evaluation Ratio (SER) via the sequential evaluation scheme, eliminating reliance on lifecycle labels.

Experimental results on the C-MAPSS dataset demonstrate that the proposed method achieves robust long-term prediction of degradation parameters, with an $R^2$ consistently above 0.8 and an RMSE around 0.1. The constructed health indicators exhibit high temporal consistency accuracy (late-stage temporal consistency metrics are around 0.8). Ablation studies further quantify the contribution of each module: the CNN module effectively extracts coupling relationships between multidimensional parameters, and its absence leads to an approximately 15% relative increase in RMSE; the MLA mechanism achieves lightweighting while maintaining accuracy; and the sequential evaluation strategy significantly enhances the monotonicity and consistency of the health index, with the late-stage consistency metric improving by over 100% compared to baseline methods. These results validate the rationality and necessity of the method's design.

Compared to various baseline models, the proposed method achieves superior prediction accuracy while significantly improving computational efficiency through lightweight network design, featuring lower parameters, computational complexity, and inference latency. This proves its feasibility for deployment in resource-constrained industrial environments (e.g., edge computing devices). Furthermore, the "parameter prediction-index fusion" framework offers better modularity and interpretability. Comparisons with Graph Neural Network models indicate that the proposed method achieves comparable or superior prediction accuracy without relying on predefined graph structures, while incurring lower computational overhead and higher inference efficiency. This demonstrates that the paradigm of "implicitly learning" internal system coupling relationships presents a practical and effective alternative in industrial scenarios where sensor physical relationships are ambiguous or where high real-time performance is required. In summary, this research provides a solution for RUL prediction of complex systems under label-scarce conditions that combines high accuracy, high efficiency, and high practicality, exhibiting strong potential for engineering application.

## Supporting information

**S1 File. 3-cnn-transformer. Implements the complete CNN-Transformer hybrid model for RUL prediction, including training and evaluation pipelines.**
(PY)

**S2 File. Dataset. Defines a PyTorch Dataset class to structure and load sensor data for batch training.**
(PY)

**S3 File. Utils. Provides utility functions for data processing, model evaluation, and result visualization.**
(PY)

## Author contributions

**Methodology:** Bo Mo.

**Software:** Feng Han.

**Supervision:** Bo Mo.

**Validation:** Feng Han.

**Writing – original draft:** Feng Han.

**Writing – review & editing:** Feng Han.

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
