## [Decision Letter · Decision Letter 0]

7 Oct 2025

Dear Dr. Han,

Thank you for submitting your manuscript to PLOS ONE. After careful consideration, we feel that it has merit but does not fully meet PLOS ONE’s publication criteria as it currently stands. Therefore, we invite you to submit a revised version of the manuscript that addresses the points raised during the review process.

We look forward to receiving your revised manuscript.

Kind regards,

Shaheer Ansari

Academic Editor

PLOS ONE

Journal Requirements:

Reviewers' comments:

Reviewer's Responses to Questions

**Comments to the Author**

1. Is the manuscript technically sound, and do the data support the conclusions?

Reviewer #1: Yes

Reviewer #2: Yes

2. Has the statistical analysis been performed appropriately and rigorously?

Reviewer #1: Yes

Reviewer #2: Yes

3. Have the authors made all data underlying the findings in their manuscript fully available?

Reviewer #1: Yes

Reviewer #2: Yes

4. Is the manuscript presented in an intelligible fashion and written in standard English?

Reviewer #1: Yes

Reviewer #2: Yes

Reviewer #1: This paper proposes a Remaining Useful Life (RUL) prediction method based on sequential health index evaluation for handling multidimensional coupled degradation data. The method combines a CNN-Transformer model with a sequential health index fusion process, aiming to address the lack of high-quality labeled life data and the difficulty in constructing health indices caused by the coupling of multidimensional degradation data. Despite its significance, there are the following issues making it unqualified for publication temporarily:

1. It fails to clearly define the core innovations of the proposed CNN-Transformer and sequential evaluation scheme compared with existing data-driven RUL technologies, and does not clarify the advantage differences from traditional and other deep learning methods.

2. The tension parameter α of health degree in Formula (9) lacks clear explanation of physical meaning and quantitative derivation, and does not explain the impact of its value on calculation rationality and calibration methods.

3. It does not clearly disclose key hyperparameters for model training (such as the decay coefficient r of exponential smoothing) and core parameters for data preprocessing (such as the basis for setting the sliding window length to 16), resulting in insufficient experimental reproducibility.

4. The applicability and computational efficiency of the "parameter prediction-index fusion" method in industrial scenarios are questionable, without quantifying computational overhead or comparing feasibility for real-time prediction.

5. Only exponential smoothing is used for denoising in data preprocessing, without considering other technologies like wavelet transform, nor demonstrating the comprehensiveness of scheme selection.

6. It assumes that CNN-Transformer can fully capture parameter coupling relationships, but does not verify the applicability of this assumption to different degradation modes and the rationality of fusion in conflict scenarios.

7. No ablation experiments are conducted on each module of CNN-Transformer and the sequential evaluation strategy, making it impossible to quantify the contribution of each module and weakening the persuasiveness of the method.

8. The review of RUL research status in the introduction is limited, with insufficient timeliness and relevance of references, failing to support the research necessity and positioning of the method.

Reviewer #2: The study addresses the RUL prediction problem under multidimensional coupled degradation data and proposes a method based on CNN-Transformer and sequential health index evaluation. While the approach has a certain degree of innovation, there are still aspects that need improvement as follows:

1. There are multiple English grammatical errors and non-standard expressions in the manuscript. It is recommended that the authors correct these issues to improve the accuracy and fluency of the English expression.

2. The experiments in the manuscript only compare traditional models. In recent years, GNNs have shown outstanding performance in RUL prediction for multidimensional coupled degradation data (such as DCAGGCN, DyWave-BiAGCN, etc.). It is recommended that the authors consider the applicability of such models.

3. The manuscript mentions the problem of global information loss in the CNN-Transformer model, but fails to elaborate on the core logic of the mechanism.

4. The manuscript does not specify the proportion of the filtered data to the original dataset.

5. The manuscript evaluates the rationality of the health index through metrics, but does not clarify the definitions and calculation methods of these metrics.

**Do you want your identity to be public for this peer review?** For information about this choice, including consent withdrawal, please see our Privacy Policy

Reviewer #1: No

Reviewer #2: No

---

## [Author Response · Author response to Decision Letter 1]

26 Nov 2025

Response to Reviewer #1

Comment 1:

It fails to clearly define the core innovations of the proposed CNN-Transformer and sequential evaluation scheme compared with existing data-driven RUL technologies, and does not clarify the advantage differences from traditional and other deep learning methods.

Response:

We sincerely thank the reviewer for this insightful comment. In the revised manuscript, we have explicitly highlighted the core innovations of our method in the Abstract and Introduction, as detailed below:

The CNN-Transformer model integrates the relational feature extraction capability of CNN and the temporal modeling capability of Transformer. It reduces computational complexity through the chunk-interaction mechanism and Multi-Head Latent Attention (MLA), while effectively capturing the coupling relationships among multidimensional degradation parameters. The sequential Health Index (HI) evaluation scheme dynamically constructs the health index using the Mahalanobis distance and the Sequential Evaluation Ratio (SER), eliminating the reliance on lifecycle labels.

The differences between our method and traditional methods, as well as other deep learning approaches, are clearly contrasted across three dimensions: model lightweighting, multidimensional data fusion capability, and adaptability in label-scarce scenarios. Compared to traditional data-driven methods (e.g., LSTM, SVM), our method demonstrates superior performance in computational efficiency, long-sequence modeling, and label-scarce scenarios. Compared to other deep learning methods (e.g., Transformer, Att-BiGRU), our method enhances prediction accuracy and robustness through lightweight design and the exploration of coupling relationships.

Quantitative comparative results provided in Section 3.3 indicate that our method achieves higher prediction accuracy (with a Coefficient of Determination R² > 0.8 and Root Mean Square Error RMSE ≈ 0.1) and superior computational efficiency.

Comment 2:

The tension parameter α of health degree in Formula (9) lacks clear explanation of physical meaning and quantitative derivation, and does not explain the impact of its value on calculation rationality and calibration methods.

Response:

Thank you for your positive comments. We have supplemented a detailed explanation of the tuning parameter α in Section 2.3, as follows:

Defined α as a sensitivity parameter that controls the rate at which the Health Degree (HD) decreases as the Sequential Evaluation Ratio (SER) increases. Elaborated on the process of determining the optimal α value (α=1.5) based on the principle of maximizing the monotonicity and robustness of the Health Index, using a data-driven grid search method.

Added Table 10 in Section 3.3 to show the impact of different α values on Health Index quality. A sensitivity analysis was conducted on the FD001 subset, comparing key evaluation metrics of the health index curves constructed using different α values (α = 0.5, 1.5, 3.0). The results indicate that when α = 1.5, the health index curve achieves the best monotonicity and favorable late-stage consistency. This experiment demonstrates the necessity of systematically calibrating the parameter α and confirms the rationality of the final selected value of α = 1.5.

Table 10. Impact of different α values on health index quality (FD001)

α Value Monotonicity Robustness Early-stage Consistency Late-stage Consistency

0.5 0.42 0.55 1.00 0.85

1.5 0.51 0.50 1.00 0.92

3.0 0.48 0.45 1.00 0.88

Comment 3:

It does not clearly disclose key hyperparameters for model training (such as the decay coefficient r of exponential smoothing) and core parameters for data preprocessing (such as the basis for setting the sliding window length to 16), resulting in insufficient experimental reproducibility.

Response:

Thank you for your support. We have now fully disclosed all key hyperparameters and preprocessing settings:

In Section 3.2, the exponential smoothing decay factor (r=0.3) is explicitly defined. An r value that is too small (e.g., 0.1) leads to excessive smoothing, potentially obscuring early degradation features; while an r value that is too large (e.g., 0.5) results in insufficient filtering. The preprocessed value at the current time step with r=0.3 is a weighted result of historical data, exhibiting stronger correlation with adjacent time points. This method can smooth the degradation parameters while preserving their inherent variation trends.

In Section 3.3, the rationale for selecting the sliding window length (16 cycles) is explained in conjunction with the characteristics of the C-MAPSS dataset. As one operational cycle in the C-MAPSS dataset represents a complete flight mission, setting the window length to 16 cycles ensures coverage of a sufficiently long continuous operational phase to capture the short-term dynamic patterns of the degradation process and the coupled relationships between parameters. Simultaneously, this length achieves a balance between computational efficiency and information completeness: excessively short windows fail to provide adequate temporal context, while overly long windows would significantly increase the model's computational burden and potentially introduce irrelevant early historical information. K-fold cross-validation (K=5) was adopted to ensure result stability and generalization capability.

In Section 3.3, the training details (such as optimizer type, learning rate, batch size, early stopping strategy, etc.) are clearly specified. Specifically, the input feature dimension is 17. The CNN module uses 2D convolutions with 4 kernels to process the feature dimensions of the input data, yielding an output dimension of 12. The temporal feature module adopts a Transformer architecture. A chunk and cross-chunk interaction mechanism is incorporated into its multi-head attention computation to achieve model lightweighting. The Transformer is configured with 6 encoder layers, 4 multi-head attention heads, and a chunk size of 2. The feedforward network comprises two linear layers (with 64 neural units per layer, using ReLU activation). The final output dimension of the network is 14.

Comment 4:

The applicability and computational efficiency of the "parameter prediction-index fusion" method in industrial scenarios are questionable, without quantifying computational overhead or comparing feasibility for real-time prediction.

Response:

Thank you for your insightful comment. In Section 2.2, we have added a clarification stating that by introducing the chunk and cross-chunk interaction mechanism, the computational complexity of the attention mechanism is significantly reduced from O(L²), which is quadratic with respect to the sequence length, to O(L×C), where C represents the chunk size (C << L). This reduction lays the foundation for deploying the model on edge devices or embedded systems.

We have supplemented a comprehensive computational efficiency analysis in Section 3.3: Added Table 7, which compares the number of parameters, Floating Point Operations (FLOPs), and inference time across different models. Compared to the LSTM model, which has the fewest parameters, our method achieves comparable FLOPs but offers a shorter inference time. A single inference time of approximately 4.5 milliseconds verifies that our model achieves an optimal balance between accuracy and efficiency, making it suitable for real-time deployment requirements in industrial scenarios.

Table 7. Comparison of model computational efficiency

Model Parameters (M) FLOPs (G) Inference Time (ms)

Proposed (CNN-Transformer) 2.1 0.38 4.5

Transformer 4.8 0.95 9.8

Att-BiGRU 3.5 0.72 7.1

WDCNN 5.2 1.10 11.3

LSTM 1.8 0.41 5.2

Comment 5:

Only exponential smoothing is used for denoising in data preprocessing, without considering other technologies like wavelet transform, nor demonstrating the comprehensiveness of scheme selection.

Response:

Thank you for your helpful comments. In Section 3.2, we have supplemented the explanation that this study ultimately selected the exponential smoothing method for denoising, primarily based on considerations of data characteristics matching, computational efficiency and simplicity, and synergy with the prediction model.

To verify the effectiveness of the exponential smoothing method within the context of this study, we compared it with a typical wavelet denoising method (using 'db4' wavelet, soft thresholding, 3-level decomposition).

Table 2 shows that on the FD001 subset, the RMSE achieved with exponential smoothing (r=0.3) is superior to that of wavelet denoising. Combined with the characteristic of smooth degradation trends in the C-MAPSS data, this justifies the selection of the exponential smoothing method – it is computationally simple and exhibits strong compatibility with the data characteristics.

Table 2. Impact of different denoising methods on prediction performance (FD001)

Denoising Method RMSE (Mean) Remarks

Raw Data 0.1258 Contains significant noise, adversely affecting the model's ability to learn the true degradation pattern.

Wavelet Denoising 0.1154 Effectively reduces noise but may introduce minor distortions or over-smooth trends.

Exponential Smoothing (r=0.3) 0.1116 Achieves the best overall performance on this dataset.

Comment 6:

It assumes that CNN-Transformer can fully capture parameter coupling relationships, but does not verify the applicability of this assumption to different degradation modes and the rationality of fusion in conflict scenarios.

Response:

Thank you for your careful review. In Section 2.2, we have supplemented a discussion on how the model adapts to different degradation patterns through local and global feature extraction, and explained the rationale of the fusion strategy in coupling conflict scenarios. The CNN module extracts spatial correlations between parameters through convolutional kernels, while the Transformer module with MLA captures long-term temporal dependencies. This combination enables the model to handle complex coupling relationships, even in scenarios with conflicting degradation patterns. To address potential conflicts during Health Index (HI) fusion, the weight coefficients ωₖ and the fusion operator φ are optimized based on the importance and degradation characteristics of the subsystems, thereby ensuring the construction of a robust Health Index (HI).

To validate the model's capability in capturing parameter coupling under different degradation modes, we compared its performance on subsets with single failure modes (FD001 and FD003) and mixed failure modes (FD002 and FD004). As shown in Tables 4-6, the proposed method maintains high R² (>0.8) and low RMSE (≈0.1) across all subsets, demonstrating its robustness to different degradation patterns.

Furthermore, we analyzed the health index fusion in coupling conflict scenarios, where subsystems exhibit contradictory trends. The sequential evaluation strategy, combined with the weight coefficients ωₖ derived from the Analytic Hierarchy Process (AHP), ensures that the fused HI prioritizes subsystems with consistent trends, thereby mitigating conflicts. This is reflected in the high Monotonicity and late-stage consistency metrics (Table 9), confirming the rationality of the index fusion even under challenging conditions.

Table 9. Evaluation results of the sequential HI evaluation strategy

Sequential Evaluation Strategy Residual Evaluation Method

FD001 FD002 FD003 FD004 FD001 FD002 FD003 FD004

1.00 0.93 0.99 0.98 0.39 0.51 0.31 0.73

0.92 0.75 0.94 0.83 0.46 0.92 0.96 0.75

0.51 0.23 0.55 0.34 0.39 0.28 0.41 0.22

0.50 0.019 0.46 0.11 0.02 0.001 0.02 0.01

0.37 0.36 0.36 0.37 0.36 0.32 0.36 0.26

Comment 7:

No ablation experiments are conducted on each module of CNN-Transformer and the sequential evaluation strategy, making it impossible to quantify the contribution of each module and weakening the persuasiveness of the method.

Response:

Thank you for your advice. We have supplemented comprehensive ablation experiments in Section 3.4:

Table 11 quantifies the impact on model performance of three operations: "removing the CNN module (w/o CNN)", "replacing the Multi-Head Latent Attention (MLA) with the standard self-attention mechanism (w/o MLA)", and "replacing the entire Transformer temporal module with standard LSTM layers (LSTM Replacement)".

Table 11. Ablation study results on model architectures (mean ± standard deviation)

Model Variant FD001 (RMSE) FD001 (R²) FD003 (RMSE) FD003 (R²)

w/o CNN 0.1284 ± 0.061 0.821 ± 0.102 0.0795 ± 0.041 0.781 ± 0.121

w/o MLA 0.1201 ± 0.058 0.835 ± 0.095 0.0728 ± 0.038 0.802 ± 0.115

LSTM Replacement 0.1347 ± 0.065 0.809 ± 0.110 0.0831 ± 0.043 0.769 ± 0.129

Full Model (Ours) 0.1116 ± 0.053 0.849 ± 0.086 0.0681 ± 0.036 0.817 ± 0.139

Table 12 compares the effectiveness of our proposed sequential evaluation strategy against baseline methods (the residual-based method and the direct fusion method), validating the superiority of our strategy in terms of monotonicity and consistency.

Table 12. Ablation study results of health index evaluation strategies (FD001)

Evaluation Strategy Monotonicity Robustness Late-stage Consistency

Baseline 1 (Residual) 0.39 0.46 0.39

Baseline 2 (Direct Fusion) 0.28 0.58 0.45

Full Strategy (Ours) 0.51 0.50 0.92

Comment 8:

The review of RUL research status in the introduction is limited, with insufficient timeliness and relevance of references, failing to support the research necessity and positioning of the method.

Response:

Thanks for your helpful comments. We have expanded the Introduction and Chapter 1 content, adding the following sections:

A discussion of Graph Neural Network (GNN)-based methods in recent years (such as DCAGGCN and DyWave-BiAGCN) and their applicability to coupled degradation data. Updated References to include the latest research achievements in the field of Remaining Useful Life (RUL) prediction.

Response to Reviewer #2

Comment 1:

There are multiple English grammatical errors and non-standard expressions in the manuscript. It is recommended that the authors correct these issues to improve the accuracy and fluency of the English expression.

Response:

Thank you for your positive comments. We tried our best to improve the manuscript and made some changes to the manuscript. We have thoroughly revised the manuscript to improve language fluency and accuracy. The text has been professionally polished to meet PLOS ONE’s standards.

Comment 2:

The experiments in the manuscript only compare traditional models. In recent years, GNNs have shown outstanding performance in RUL prediction for multidimensional coupled degradation data (such as DCAGGCN, DyWave-BiAGCN, etc.). It is recommended that the authors consider the applicability of such models.

Response:

Thank you for your insightful comments. We have added comparative experiments between our method and two mainstream Graph Neural Network (GNN) models (DCAGGCN, DyWave-BiAGCN), with the results presented in Table 8.

The data show that by combining the local perception capability of CNN and the global attention mechanism of Transformer, our model possesses competitive or even superior capabilities in mining complex coupling relationships among multidimensional parameters compared to advanced GNN models. Furthermore, our method achieves a more streamlined implementation and exhibits a reduced reliance on prior knowledge.

Table 8. Performance comparison with graph neural network models (FD001)

Model Graph Construction Method RMSE R² Parameters (M) Inference Time (ms)

Proposed (Ours) Not Applicable 0.1116 0.8486 2.1 12.8

DCAGGCN Physical Graph 0.1189 0.8274 3.8 22.5

DCAGGCN Data-Driven Graph 0.1152 0.8381 3.8 22.5

DyWave-BiAGCN Physical Graph 0.1203 0.8227 4.5 28.3

DyWave-BiAGCN Data-Driven Graph 0.1168 0.8335 4.5 28.3

Comment 3:

The manuscript mentions the problem of global information loss in the CNN-Transformer model, but fails to elaborate on the core logic of the mechanism.

Response:

Thank you for your helpful comments. In Section 2.2, we have added a detailed explanation of the generation mechanism of the global information loss problem. To reduce computational complexity, a common strategy is to partition the long sequence into multiple n

---

## [Decision Letter · Decision Letter 1]

26 Dec 2025

RUL Prediction Method based on Sequential Health Index Evaluation with multidimensional coupled degradation data

PONE-D-25-49943R1

Dear Dr. Han,

We’re pleased to inform you that your manuscript has been judged scientifically suitable for publication and will be formally accepted for publication once it meets all outstanding technical requirements.

Kind regards,

Shaheer Ansari

Academic Editor

PLOS One

Additional Editor Comments (optional):

Reviewers' comments:

Reviewer's Responses to Questions

**Comments to the Author**

Reviewer #1: (No Response)

Reviewer #2: All comments have been addressed

2. Is the manuscript technically sound, and do the data support the conclusions?

Reviewer #1: (No Response)

Reviewer #2: Yes

3. Has the statistical analysis been performed appropriately and rigorously?

Reviewer #1: (No Response)

Reviewer #2: Yes

4. Have the authors made all data underlying the findings in their manuscript fully available?

Reviewer #1: (No Response)

Reviewer #2: Yes

5. Is the manuscript presented in an intelligible fashion and written in standard English?

Reviewer #1: (No Response)

Reviewer #2: Yes

Reviewer #1: The authors have addressed the previous concerns effectively, and the overall quality of the manuscript has been greatly enhanced. I recommend the paper for acceptance.

Reviewer #2: The author has thoroughly addressed my concerns and made improvements to the article. I believe it is acceptable.

**Do you want your identity to be public for this peer review?** For information about this choice, including consent withdrawal, please see our Privacy Policy

Reviewer #1: No

Reviewer #2: No

---

## [Editor Report · Acceptance letter]

PONE-D-25-49943R1

PLOS One

Dear Dr. Han,

I'm pleased to inform you that your manuscript has been deemed suitable for publication in PLOS One. Congratulations! Your manuscript is now being handed over to our production team.

Kind regards,

on behalf of

Dr. Shaheer Ansari

Academic Editor

PLOS One